# Leptin in the Commissural Nucleus of the Tractus Solitarius (cNTS) and Anoxic Stimulus in the Carotid Body Chemoreceptors Increases cNTS Leptin Signaling Receptor and Brain Glucose Retention in Rats

**DOI:** 10.3390/medicina58040550

**Published:** 2022-04-16

**Authors:** Mónica Lemus, Cynthia Mojarro, Sergio Montero, Mario Ramírez-Flores, José Torres-Magallanes, Adrián Maturano-Melgoza, Elena Roces de Álvarez-Buylla

**Affiliations:** 1Neuroendocrinology Department, University Center Biomedical Research, Colima University, Colima 28045, Mexico; mlv@ucol.mx (M.L.); cynxi@icloud.com (C.M.); ramirez@ucol.mx (M.R.-F.); flipy_joan@hotmail.com (J.T.-M.); 2Medicine Faculty, Colima University, Colima 28040, Mexico; 3Nursing Faculty, Colima University, Colima 28040, Mexico; maturano_melgoza@ucol.mx

**Keywords:** brain glucose retention, carotid body chemoreceptors, *c-Fos*, commissural nucleus tractus solitarius, leptin, *Ob-Rb* genes

## Abstract

*Background and Objectives*: The commissural nucleus of the tractus solitarius (cNTS) not only responds to glucose levels directly, but also receives afferent signals from the liver, and from the carotid chemoreceptors (CChR). In addition, leptin, through its receptors in the cNTS, regulates food intake, body weight, blood glucose levels, and brain glucose retention (BGR). These leptin effects on cNTS are thought to be mediated through the sympathetic–adrenal system. How these different sources of information converging in the NTS regulate blood glucose levels and brain glucose retention remains largely unknown. The goal of the present study was to determine whether the local administration of leptin in cNTS alone, or after local anoxic stimulation using sodium cyanide (NaCN) in the carotid sinus, modifies the expression of leptin *Ob-Rb* and of *c-Fos* mRNA. We also investigated how leptin, alone, or in combination with carotid sinus stimulation, affected brain glucose retention. *Materials and Methods*: The experiments were carried out in anesthetized male Wistar rats artificially ventilated to maintain homeostatic values for pO_2_, pCO_2_, and pH. We had four groups: (a) experimental 1, leptin infusion in cNTS and NaCN in the isolated carotid sinus (ICS; *n* = 10); (b) experimental 2, leptin infusion in cNTS and saline in the ICS (*n* = 10); (c) control 1, artificial cerebrospinal fluid (aCSF) in cNTS and NaCN in the ICS (*n* = 10); (d) control 2, aCSF in cNTS and saline in the ICS (*n* = 10). *Results*: Leptin in cNTS, preceded by NaCN in the ICS increased BGR and leptin *Ob-Rb* mRNA receptor expression, with no significant increases in *c-Fos* mRNA in the NTSc. *Conclusions*: Leptin in the cNTS enhances brain glucose retention induced by an anoxic stimulus in the carotid chemoreceptors, through an increase in *Ob-Rb* receptors, without persistent changes in neuronal activation.

## 1. Introduction

Leptin, a 16 kD protein produced mostly by adipocytes, is a key hormone in the regulation of body weight [1]. Leptin inhibits food intake and increases energy expenditure [2] through the modulation of several hypothalamic neuropeptides; inhibiting neuropeptide Y (NPY) and agouti-related protein (AgRP), and stimulating cocaine- and amphetamine-regulated transcript (CART), proopiomelanocortin (POMC) [3]. Importantly, leptin also participates in blood glucose regulation [4]. For example, leptin injection in a non-obese rodent model of type 1 diabetes, results in the normalization of blood glucose [5].

Elevated leptin level increases hypothalamus signaling to decrease adiposity [4], but also suppresses glucagon and corticosterone production, elevation of glucose uptake, and inhibition of hepatic glucose output [6]. However, the central mechanisms of glucose regulation by leptin remain poorly understood.

Leptin receptors are widely expressed in the central nervous system (CNS) [7,8,9]. These receptors are present in the hypothalamus in high levels, and in the brainstem commissural nucleus of the tractus solitarius (cNTS) [9]. Specifically, the caudal NTS (cNTS) [10] and the medial gelatinous NTS [11] express high levels of the long isoform of the leptin signaling receptor (*Ob-Rb*) [8]. The cNTS also receives primary afferents from the carotid body chemoreceptors (CChR) [12]. In addition to pO_2_ and pH, cells in the CChR are sensitive to glucose levels and participate in glucose homeostasis [13,14]. A decrease in carotid blood glucose levels results in increased CChR activity [15]. Anoxic stimulation of the CChR induces systemic hyperglycemia and an increase in brain glucose retention (BGR) [16]. In addition, the cNTS receives afferents from glucose-sensing regions of the hypothalamus [17] and the liver [18]. Furthermore, glucose-sensitive neurons have also been suggested to exist within the cNTS [19]. Therefore, key peripheral and central information on systemic blood glucose levels converge in the cNTS raising the question of how this information on glucose levels is integrated with local leptin signaling. Microinjections of leptin into the NTS in rats exposed to chronic-intermittent hypoxia increases sympathetic activity resulting in increased heart activity and blood pressure [20]. A previous study from our laboratory has shown that the local administration of leptin into the cNTS potentiates systemic hyperglycemia and BGR induced by the stimulation of the CChR [16].

In the present study, we investigated the effects of leptin, locally applied to cNTS on the local levels of *Ob-Rb* mRNA [5] and *c-Fos* expression in this nucleus. The latter as a proxy for general neuronal activation. In order to determine how leptin effects in cNTS were integrated when CChR receptors were activated, we also studied the effects of leptin in NTS in the presence of local anoxic CChR stimulation. Our results indicate that leptin in the cNTS potentiates BGR induced by an anoxic stimulus in the CChR through a local increase in leptin *Ob-Rb* gene expression.

## 2. Materials and Methods

### 2.1. Ethics

All experimental procedures with rats were approved by the University of Colima Committee (No. 2014–02) on the Use and Care of Animals in Accordance with Association for Assessment and Accreditation of Laboratory Animal Care from National Institutes of Health Guidelines, USA, 2010 [21].

### 2.2. Animals and General Surgery

The experimental procedures were performed in male Wistar rats weighing 270 to 300 g, maintained individually in small acrylic cages with 12 light/dark cycles, and with free access to food and water with 10% glucose. Food was withdrawn 12 h before surgery. The rats were anesthetized with intraperitoneal (i.p.) sodium pentobarbital injection (3.3 mg/100 g saline, Pfizer, Ciudad de México, México), and maintained with the same anesthetic (0.063 mg/min in Sal). Anesthesia was monitored by means of palpebral and leg prick reflexes. Buprenorfin (0.03 mg/kg s.c.) was used as an analgesic. Rat body temperature was maintained at 37 °C with a thermal cushion (For Paws Nature Heat, Hauppauge, NY 1178, USA). The rats were artificially ventilated (ventilator Ugo Basile, Stoelting, Wood Dale, IL, USA), through a tracheal cannula to adjust velocity and volume and to maintain the homeostatic values (pO_2_, pCO_2_, and pH) [16].

### 2.3. Experimental Protocol

Forty healthy anesthetized rats were used to measure *Ob-Rb* gene expression and *c-Fos* in the cNTS. We also measured BGR in all these animals as previously described [16]. Rats were allowed to stabilize for 30 min after surgery. Rats were randomly assigned into two experimental and two control groups: (a) Experimental 1: Leptin in cNTS and NaCN in ICS (*n* = 10); (b) Experimental 2: Leptin in cNTS and Sal. in ICS (*n* = 10); (c) Control 1: aCSF in cNTS and NaCN in the ICS (*n* = 10); (d) Control 2: aCSF in cNTS and Sal. in ICS (*n* = 10). Five rats for each group were utilized for histological analysis and five for RT-PCR biochemical analysis.

### 2.4. Materials and Drugs

Buprenorphine (Schering Plough, Ciudad de México, México, Temgesic, 0.005 mg/100 g i. muscular) [22]; cerebrospinal fluid (aCSF, NaCl 145 mM, KCl 2.7 mM, MgCl_2_ 1.0 mM, CaCl_2_ 1.2 mM, ascorbic acid, C_6_H_8_O_6_ 0.2 mM, NaH_2_PO_4_ 2.0 mM, in 100 mL tri distilled water with a pH of 7.3–7.4) (20 nL) [23]; leptin (Sigma, San Pedro Garza García, México) (50 ng/20 nL in aCSF); saline (Pisa, Ciudad de México, México) (0.9% 50 nL) [13]; sodium cyanide (NaCN) (Fluka Biochemika, Sigma-Aldrich, Naucalpan de Juárez, México 5 µg/100 g/50 nL saline) [13]; sodium pentobarbital (Pfizer, México) (3 mg/100 g mL saline) [13]; methylene blue (Sigma-Aldrich, México) (50 nL en LCRa).

### 2.5. Isolated Carotid Sinus Preparation, Carotid Chemoreceptors Stimulation and BGR Measurement

The CS was isolated from systemic and cerebral circulations for 15 to 20 s, during the time of application of NaCN. A polyethylene catheter (Clay Adams PE-10: 0.28 mm ID, 0.61 mm OD; Parsippany, NJ, USA) filled with heparin (1000 U/mL) was inserted via the lingual artery to the carotid (5 mm before the carotid bifurcation). NaCN (5 µg/100 g BW) or saline in 50 nL were injected through this catheter to the isolated carotid sinus. Any locally remaining NaCN was washed through the same syringe, before re-establishing normal circulation to the CS. To determine the BGR, two catheters were inserted, one into the femoral artery, and the another one in the jugular sinus from the external jugular vein. The correct placement of all catheters was verified at the end of each experiment [13], by a neck skin incision to visualize the confluence of the external and internal jugular veins and the tip of the catheter.

### 2.6. Microinjections of Drugs in the cNTS

Saline or leptin were applied to the cNTS using a stereotaxic frame (Stoelting Co., Wood Dale, IL, USA) and the following coordinates from lambda: AP = −5.1 mm, L = 0.1 mm, dorsal V = 8.1 mm, from Paxinos and Watson [24]. The surface of the skull was cleaned with hydrogen peroxide and a small burr (1.6 mm approx.) hole was made on the skull. A pulled glass micropipette (40–50 µm external tip diameter, Drummond Scientific Co., Broomall, PA, USA) filled with leptin or aCSF was slowly lowered into the cNTS. The micropipette was connected to a 0.5 µL micro-syringe (Hamilton, Reno, NV, USA) with silastic tubing (PE 20) (Clay Adams, Persippany, NJ, USA) for injections [25]. Leptin (50 ng/20 nL aCSF) or the same volume of aCSF were injected into the left cNTS 4 min before NaCN or saline (Sal) applied to the isolated left carotid sinus. At the end of each experiment, the site of infusion was marked by injecting the same volume of methylene blue (50 nL in aCSF) using the same cannula [26].

After the injections and experimental measurements, the rats were removed from the stereotaxic instruments and euthanized by decapitation [27] under anesthesia. The brain was rapidly extracted and frozen at −70 °C in a freezer (Revco) and the brain was sectioned coronally into 40 µm in a cryostat (Leica CM1800, Leica Microsystems, Nusloch, Germany). The sections were mounted and stained with cresyl violet at 0.5% [24], they were covered with Entellan 8 (Merck, Darmstadt, Germany) to be analyzed in a dissecting microscope and photographed (Zeiss AxioCam Carl Zeiss, Munich, Germany) (Figure 1). In 4 out of 44 rats, the injections missed the cNTS and these rats were eliminated from our analysis.

### 2.7. Biochemical Assays

Glucose concentration in blood was measured with a digital glucometer (ACCU-CHEK Performa, Roche, Ciudad de México, México) in µmol/mL. One hundred microliters of arterial blood (0.1 mL) were obtained with a catheter inserted in the femoral artery. The same volume of venous blood (0.1 mL) was obtained from the jugular sinus by a catheter placed in the external jugular vein. To determine the basal levels of blood glucose, two samples (separated by 5 min interval) of arterial and venous blood were obtained for each rat. The values were averaged to obtain the basal glucose levels. After the NaCN injection or Sal (5 µg/100 g) in the isolated carotid sinus (t = 0), four experimental samples were obtained at t = 5 min, t = 10 min, t = 20 min, and t = 30 min. The volume of blood obtained was less than 5% of the total circulating volume in the rat [28]. Carotid blood flow was monitored without altering blood circulation, using a Doppler ultrasonic meter placed around the right carotid artery, and was registered in mL/g/min [29,30]. Glucose retention by the brain was calculated in µmol/g/min multiplying the differences between arterial and venous glucose concentrations by arterial blood flow in the carotid [13].

### 2.8. RT-PCR Analysis of Leptin and c-Fos mRNA Genes in the cNTS

This analysis was based on a method described by Lainez and Coss, 2019 [31] with some modifications. The brain was rapidly extracted and sliced on a cooled (4 °C) Brain Matrix (Bioanalytical Systems Inc., West Lafayette, IN, USA). The indicated 1 mm hind-brain slab containing the NTS was placed on a cooling plate and the regions of the NTS were microdissected (~100 mg of tissue was obtained) (Figure 2). The dissected NTS was placed in 1 mL of Trizol reagent (Ambion by Thermo Fisher Scientific^TM^, Carlsbad, CA, USA) and frozen at −70 °C until use. The samples were homogenized with an ultrasonic processor (Cole-Parmer, Mod. GEX130, Vernon Hills, IL, USA). RNA was extracted in Trizol Reagent following the recommended manufacturer procedure. cDNA was prepared using a SuperScript^TM^ III First-Strand Synthesis SuperMix kit (Invitrogen by Thermo Fisher Scientific^TM^, Carlsbad, CA, USA), using the manufacturer primer oligo (dT) 20 synthesis option.

### 2.9. Genes Expression Quantification

Real-time PCR was used to determine the level of gene expression using LightCycler FastStar DNA MasterPlus SyBR Green kit (Roche, Indianapolis, IN, USA) in 20 µL containing 1× Master Mix (FastStat Taq DNA polymerase, reaction buffer, dNPT mix, SYBR Green I dye and MgCl_2_), 0.5 µM of each primer, and 10 µL of cDNA. The primers used *Ob-Rb* were: Forward-5′ AT GAAGTGGCTTAGAATCCCTTCG-3′ and reverse-5′ ATATCACTGATTCTGCATGCT-3′ [32]. For *c-Fos* forward 5′-GGGATAGCCTCTCTTACTACCACT-3′ and reverse-5′-GGGGTGTTGAAGGTCTCAAA-3 [33]. Amplification and detection were conducted with an Eppendorf MasterCycler Real lPlex as follows: pre-incubation (95 °C, for 10 min); amplification and quantification (45 cycles at 95 °C—10 s each, 58 °C—15 s each, 72 °C—10 s each); melting curve 95 °C 0 s, 65 °C 15 s, 95 °C, 0 s (with temperature ramp of 0.1 °C, fluorescence continuous detection) and cooling: 1 cycle, 4 °C.

### 2.10. Data Analysis

Statistic values for glucose retention are expressed as means ± standard deviation of the mean (SDM). All data were analyzed using the SPSS 12.0 software (IBM Co., Armonk, NY, USA) on PC computers. Statistical significance differences between groups were determined using analysis of variance (ANOVA) for multiple comparisons with Scheffé post hoc test. Quantification of *c-Fos* mRNA and *Ob-Rb* mRNA with RT-PCR was conducted with ΔΔCT software [34] and with realplex sequences, 2.2 version with raw data archives. The Program of Applied Biosystems (AB applied Biosystem, Foster City, CA, USA) was used to calculate the umbral cycle (Ct).

Significance was set at *p* < 0.05. For semi-quantitative target gene expression by RT-PCR, the ratio of the target gene to the house-keeping gene-actin was used as arbitrary units that were standardized.

## 3. Results

### 3.1. Brain Glucose Retention

Stimulation of the ICS with NaCN increases BGR [13]. We have previously reported that leptin locally applied into NTS potentiates the effects that ICS stimulation has on BGR [16]. A similar enhancement of BGR was observed in the present set of rats. Control injection of aCSF into cNTS followed by saline into the ICS (aCSF + Sal, *n* = 10) had no effect on BGR at any of the times tested. Similarly, no significant effects on BGR were observed when leptin was microinjected into NTS followed by Sal in ICS (leptin + Sal and *n* = 10) (Figure 3).

The local application of NaCN to the ICS, in animals that received aCSF in cNTS (aCSF + NaCN, *n* = 10) resulted in a significant increase (*p* < 0.05), compared with their basal values, in BGR at all time points studied. A maximum increase was observed at t = 10 min post injection from 0.32 ± 0.08 to 1.3 ± 0.09 µmol/g/min (Figure 3). This stimulation in BGR by NaCN was significantly enhanced (*p* < 0.001) by the local microinjection of leptin into cNTS (leptin + NaCN, *n* = 10), at all times studied.

These observations indicate that leptin in cNTS enhances BGR induced by ICS stimulation. Control rats group leptin + Sal (*n* = 10), that received Sal in the ICS and leptin in the cNTS, did not show a significant change in BGR (Figure 3). However, when separate groups were compared with their respective times, this increase was significant only when leptin + NaCN was compared with aCSF + NaCN at t = 10 min and 30 min (*p* = 0.044 and *p* = 0.038, respectively).

### 3.2. Leptin Ob-Rb mRNA Relative Expression in cNTS

Previous work has shown that the *Ob-Rb* receptors, measured by qPCR, are present in neurons of the cNTS [7]. NaCN injections in the ICS, leptin microinjections in cNTS or both could alter the expression of these receptors in cNTS. We therefore measured the levels of Ob-RB mRNA by qPCR in the microdissected cNTS in controls and the different experimental groups described above (Figure 3). We observed an increase in the relative OB-Rb levels in cNTS after NaCN stimulation (aCSF + NaCN, *n* = 5) or after leptin in cNTS (leptin + Sal = 5), however, this increase did not reach statistical significance (*p* = 0.20). However, when leptin was microinjected into the cNTS followed by NaCN stimulation of the ICS (leptin + NaCN, *n* = 5), an eight-fold significant (*p* = 0.03) increase in the relative levels of cNTS OB-Rb was observed compared to controls (aCSF + Sal = 5) (Figure 4). The above results indicate that leptin in cNTS or NaCN stimulation of the ICS, both may have a small effect in the expression of leptin *Ob-Rb* receptors; but when these two stimuli are combined, the expression of these receptors is clearly increased.

### 3.3. Relative Expression of c-Fos mRNA Gene Levels

Increase in the expression of the immediate early gene *c-Fos* is associated with increased neuronal activity. In order to determine if the microinjection of leptin in cNTS, the stimulation of the ICS with NaCN, or both, alter *c-Fos* expression in cNTS, we measured local levels of *c-Fos* mRNA by qPCR. Compared to aCSF + Sal controls, all other groups leptin + NaCN, leptin + Sal and aCSF + NaCN, (*p* = 0.22, *p* = 0.46 and *p* = 0.19, respectively) showed an increase in *c-Fos* mRNA levels, but this increase did not reach statistical significance. *c-Fos* mRNA levels in cNTS were very similar between aCSF + NaCN, leptin + Sal or leptin + NaCN suggesting that the combined stimulation of cNTS does not result in persistent increased neuronal activity compared to ICS stimulation alone. Leptin + NaCN, exogenous leptin, or aCSF in the cNTS, 4 min before NaCN injection or saline in the ICS did not show significant increases in *c-Fos* gene expression 35 min after the stimulation, although there was a tendency to increase this gene expression, principally in the groups that received leptin in the cNTS (leptin + Saline or leptin + NaCN) (Figure 5). Since the expression of *c-Fos* mRNA was measured 35 min after the cNTS injections, we cannot discard that transient changes in the transcription of *c-Fos*, or in neuronal activity, were not detected. Our findings suggest that following leptin injection into cNTS, NaCN stimulation of the ICS, or both, there is no persistent neuronal activation in the region of cNTS.

## 4. Discussion

Here we show that the local anoxic stimulus to the ICS, preceded by microinjection of leptin in the cNTS, increases BGR (Figure 3) and *Ob-Rb* mRNA expression (Figure 4). In contrast, we did not observe a significant change in *c-Fos* mRNA expression in cNTS (after 35 min) using the above combined or single stimulations with leptin in cNTS and NaCN in ICS (Figure 5). Our results suggest that leptin in the cNTS potentiates the BGR induced by an anoxic stimulus in the carotid bodies. This effect is associated with increased expression of the mRNA for the *Ob-Rb* receptor with no persistent changes in cNTS neural activity. Given the importance of the NTS in the cardiovascular regulation, it is possible that the increase BGR we observed was due to changes in encephalic blood volume due to changes in arterial versus venous flow. However, for our calculations of BGR, we measured blood flow in the carotid artery and saw no significant change in brain blood flow. Therefore, we infer that changes in BGR were due to differences between arterial and venous glucose concentrations.

With leptin, the NTS appears to be involved in multiple homeostatic responses, including the regulation of glucose levels, food intake, respiration, and heart rate [16,20,35,36,37,38]. The importance of leptin in glucose homeostasis has been highlighted by multiple studies (reviewed in D’souza, 2017 [6]). For example, the systemic injection of leptin in a non-obese rodent model of insulin-deficient type 1 diabetes, results in normalization of glycemia following a glucose challenge [5]. Understanding how leptin modulates glucose homeostasis is therefore an important goal in trying to treat metabolic disorders. The commissural nucleus of the NTS can respond to glucose directly, or through hepatic–portal glucose-sensitive afferents [18]. NTS also receives afferents from CChR [9,12], which has been shown to have glucose-sensing properties [13,15]. Since the NTS also has high levels of leptin (*Ob-Rb*) receptors [7,9,35,36,37,38], it is likely a key nucleus in the coordinated regulation of food intake and circulating levels of glucose.

The microinjections of leptin in the caudal NTS increase blood pressure likely due to sympathetic activation [20]. A recent study also indicates that infusion of leptin in NTS for 24 h lowers the threshold for leptin effects on food intake in the hypothalamus [39]. These effects appear to be mediated through neural connections between the NTS and the hypothalamus [39]. In our study, the acute injection of leptin in cNTS did not result in a significant increase in *c-Fos* expression in cNTS, suggesting that the effects of leptin observed are not due to the persistent neural activation in cNTS. However, it is possible that the long-term exposure for 24 h to leptin in NTS results in *c-Fos* activation. We also cannot exclude that following an acute injection of leptin, the neural activation is transient, and this is not manifested as a persistent elevation in *c-Fos* levels. Following leptin stimulation in NTS, STAT-3 activation can occur in a subpopulation of NTS neurons in the absence of *c-Fos* up-regulation [40]. It will be interesting to investigate whether the potentiation in BGR we observed when leptin is locally applied to NTS is dependent on STAT-3 activation.

In our experiments, the injection of leptin alone in the NTS did not affect BGR, suggesting that the observed potentiation is not simply due to this hormone having additive effects. Instead, the local effects of leptin in NTS on glucose regulation seem to be context dependent. It has been reported that leptin in NTS induces in a dose-dependent manner, an increase in pulmonary ventilation and an increase of the respiratory volume [38]. NTS is likely important in the integration of postprandial state, through leptin, with levels of ventilation and the circulating levels of glucose.

Intracerebroventricular injection of leptin has also been shown to promote hepatic gluconeogenesis [41], but only with reduced levels of insulin induced by sympathetic nervous system activation [42]. It has been previously shown that the NTS contains a high concentration of leptin Ob-Rb receptors [7,35,36,37,38]. Interestingly, in our results, NaCN in the carotid body 4 min after leptin microinjection in the cNTS induced a significant increase in Ob-Rb mRNA relative expression after as little as 30 min [43] (Figure 4). Leptin receptors have been also observed in a subpopulation of glial fibrillary acid protein positive astrocytes in NTS [44]. We cannot exclude that the receptors detected in our study were those in glial cells and that the observed effect of leptin in NTS was mediated by these astrocytes.

The observed potentiation in BGR when leptin was locally infused into the NTS suggests that it is within this nucleus that integration between leptin levels and glucose regulation occurs. It has been reported that leptin increasing the sensitivity of LepR-expressing neurons in the NTS, could potentiate gastrointestinal satiation signals [45]. We did not study cholecystokinin (CCK) or adenosine monophosphate-activated protein kinase (AMPK), but it will be interesting to determine how these signaling pathways are modified following leptin injection into the NTS or NaCN stimulation of the isolated carotid sinus. Indeed, Akieda-Asai et al. found that co-injection of CCK and leptin reduces food intake and this is mediated by reduced AMPK phosphorylation [46]. It has been reported that the chronic hyperphagia in knockdown of LepR in mNTS and area postrema (AP) neurons of rats (LepRKD) is likely mediated by a reduction in leptin potentiation of gastrointestinal satiation signaling, as LepRKD rats showed decreased sensitivity to the intake-reducing effects of cholecystokinin. LepRKD rats showed increased basal AMPK activity in mNTS/AP [7].

## 5. Conclusions

We conclude that leptin in the cNTS potentiates brain glucose retention induced by an anoxic stimulus in the isolated carotid sinus through an increase in leptin *Ob-Rb* gene expression without a persistent upregulation of *c-Fos* gene expression in the cNTS.

## Figures and Tables

**Figure 1 medicina-58-00550-f001:**
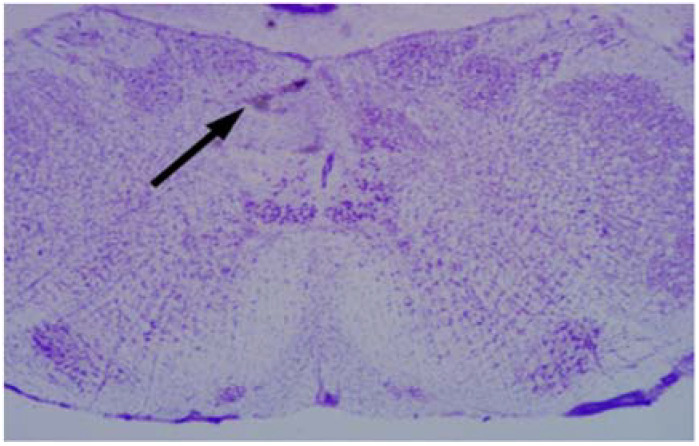
Brain stem showing the injection site in the commissural nucleus tractus solitarius.

**Figure 2 medicina-58-00550-f002:**
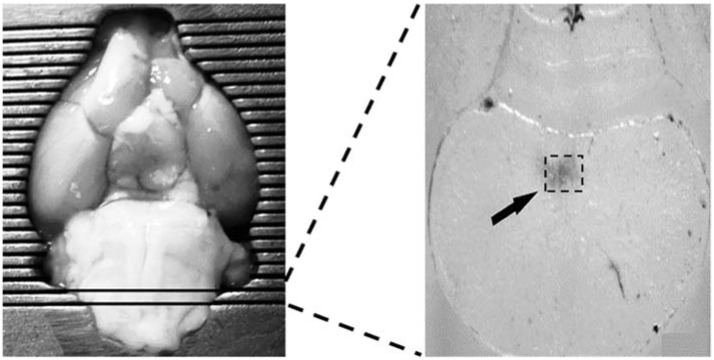
Dissection procedure for the NTS. The brain was sliced using a brain matrix and the cNTS microdissected from the indicated level as shown in the right panel.

**Figure 3 medicina-58-00550-f003:**
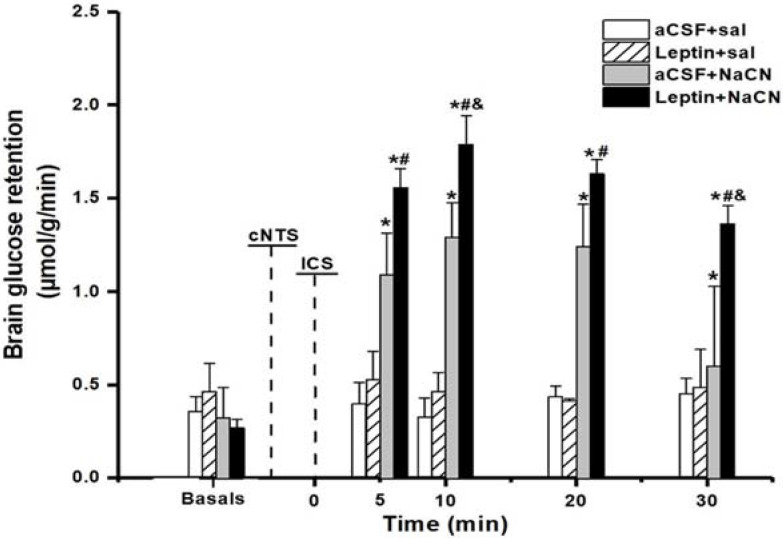
Rat brain glucose retention after leptin or control aCSF injections in the cNTS with NaCN or saline (Sal) in the ICS. The values are means ± SDM. * *p* < 0.05 vs. basal values; ^#^
*p* < 0.05 vs. leptin + saline; ^&^
*p* < 0.05 vs. aCSF + NaCN. ANOVA with Scheffé post hoc test. cNTS, injection into the commissural nucleus tractus solitarius of leptin or aCSF; ICS, injection into the isolated carotid sinus of NaCN or saline. aCSF, artificial cerebrospinal fluid.

**Figure 4 medicina-58-00550-f004:**
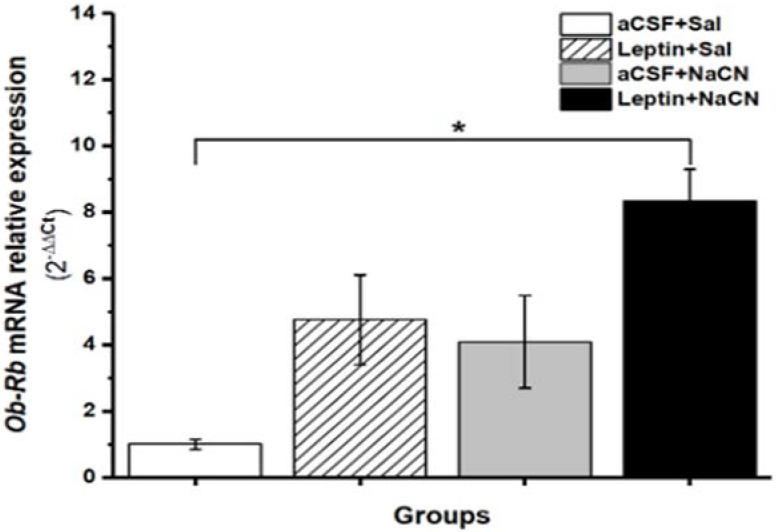
Relative expression of leptin *Ob-Rb* mRNA receptors in control (aCSF + Sal), or following leptin in cNTS (leptin + Sal), NaCN in the ICS (aCSF + NaCN), or both leptin in cNTS and NaCN in the ICS (leptin + NaCN). The values are means ± SDM. * *p* = 0.03. ANOVA with Scheffé post hoc test. aCSF, artificial cerebrospinal fluid; cNTS, commissural nucleus tractus solitarius; isolated carotid sinus (IC); NaCN, sodium cyanide; Sal, saline.

**Figure 5 medicina-58-00550-f005:**
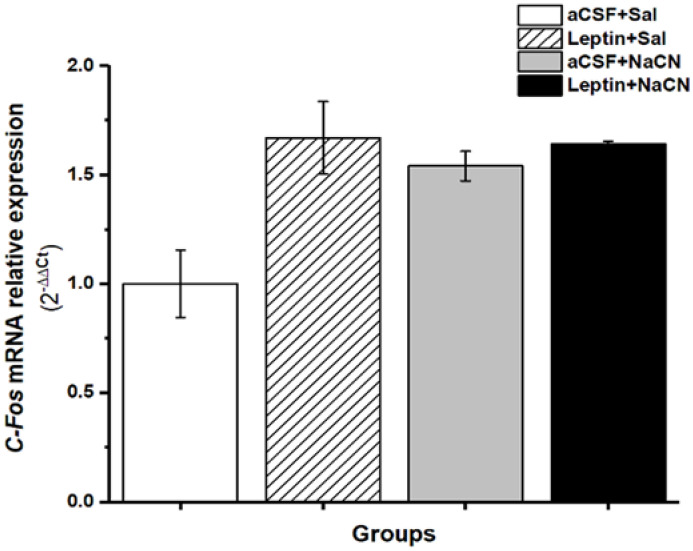
Relative expression of *c-Fos* mRNA receptors in healthy rats with leptin or aCSF injections in the cNTS before NaCN or saline the isolated carotid sinus (ICS). The values are the mean ± SDM. To calculate the umbral cycle (Ct) the software ΔΔC_T._ method was used. ANOVA with Scheffé post hoc test. aCSF, artificial cerebrospinal fluid; cNTS, commissural nucleus tractus solitarius; NaCN, sodium cyanide; Sal, saline.

## Data Availability

The data presented in this study are available upon request from the corresponding author.

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
