# Peer review of "Leptin in the Commissural Nucleus of the Tractus Solitarius (cNTS) and Anoxic Stimulus in the Carotid Body Chemoreceptors Increases cNTS Leptin Signaling Receptor and Brain Glucose Retention in Rats"

_medicina, 2022, doi:10.3390/medicina58040550_

Round 1

Reviewer 1 Report

In the present manuscript entitled “Leptin in the commissural nucleus of the tractus solitarius  (cNTS) and anoxic stimulus in the carotid body chemoreceptors increases cNTS leptin signaling receptor and brain glucose retention in rats” the authors confirmed that leptin microinjection into the cNTS with local anoxic stimulation in the carotid sinus increased mRNA expression of leptin Ob-Rb but not of C-fos. However, some concerns should be addressed.

Major comments:

1) Leptin Ob-Rb mRNA expression in cNTS was increased after Leptin injection (4 min) and NaCN in isolated left carotid sinus (30 min). How can it be explained that mRNA expression was changed for a short time (just minutes)? Normally mRNA will be altered after hours.

2) Is there any correlation to these experimental findings in Leptin signaling? Such as cholecystokinin (CCK) or adenosine monophosphate-activated protein kinase (AMPK).

Minor comments:

Line 40-41: The spelled-out form of name, not abbreviation, should be used for the first time.

Line 43: “Elevated leptin levels increase“should be “Elevated leptin level increases”

Author Response

In the present manuscript entitled “Leptin in the commissural nucleus of the tractus solitarius  (cNTS) and anoxic stimulus in the carotid body chemoreceptors increases cNTS leptin signaling receptor and brain glucose retention in rats” the authors confirmed that leptin microinjection into the cNTS with local anoxic stimulation in the carotid sinus increased mRNA expression of leptin Ob-Rb but not of C-fos. However, some concerns should be addressed.

 Major comments:

Point 1: Leptin Ob-Rb mRNA expression in cNTS was increased after Leptin injection (4 min) and NaCN in isolated left carotid sinus (30 min). How can it be explained that mRNA expression was changed for a short time (just minutes)? Normally mRNA will be altered after hours.

RESPONSE 1: It will be interesting to determine the precise time-course of mRNA changes.  At the 30 min interval used in our study we detected a significant increase in Ob-Rb mRNA expression.  This is not unprecedented;  for example:  Maroni et al. find that with acute leptin treatment the mRNA for the long isoform of leptin receptor (OB-Rb) is increased in C2C12 myotubes after as little as 30min, without affecting the levels of the short isoform (OB-Ra).

Maroni P, Citterio L, Piccoletti R, Bendinelli P. Sam68 and ERKs regulate leptin-induced expression of OB-Rb mRNA in C2C12 myotubes. Mol Cell Endocrinol. 2009 Oct 15;309(1-2):26-31. doi: 10.1016/j.mce.2009.05.021. Epub 2009 Jun 11. PMID: 19524014.

Point 2: Is there any correlation to these experimental findings in Leptin signaling? Such as cholecystokinin (CCK) or adenosine monophosphate-activated protein kinase (AMPK).

RESPONSE 2:  We did not study CCK or AMPK, but it will be interesting to determine how these signaling pathways are modified following Leptin injection into the NTS or NaCN stimulation of the isolated carotid sinus.  Indeed, Akieda-Asai found that co-injection of CCK and leptin reduces food intake and this is mediated by reduced AMPK phosphorylation.  

Akieda-Asai S, Poleni PE, Date Y. Coinjection of CCK and leptin reduces food intake via increased CART/TRH and reduced AMPK phosphorylation in the hypothalamus. Am J Physiol Endocrinol Metab. 2014 Jun 1;306(11):E1284-91. doi: 10.1152/ajpendo.00664.2013. Epub 2014 Apr 15. PMID: 24735891.

The group of Harvey J Grill signaling that The chronic hyperphagia of mNTS/AP LepRKD rats is likely mediated by a reduction in leptin potentiation of gastrointestinal satiation signaling, as LepRKD rats showed decreased sensitivity to the intake-reducing effects of cholecystokinin. LepRKD rats showed increased basal AMP-kinase activity in mNTS/AP micropunches, and pharmacological data suggest that this increase provides a likely mechanism for their chronic hyperphagia. Overall these findings demonstrate that LepRs in mNTS and AP neurons are required for normal energy balance control.

Hayes MR, Skibicka KP, Leichner TM, et al. Endogenous leptin signaling in the caudal nucleus tractus solitarius and area postrema is required for energy balance regulation [published correction appears in Cell Metab. 2016 Apr 12;23(4):744]. Cell Metab. 2010;11(1):77-83. doi:10.1016/j.cmet.2009.10.009

Minor comments:

Point 3:  Line 40-41: The spelled-out form of name, not abbreviation, should be used for the first time.

RESPONSE 3: These names are now spelled-out: neuropeptide Y (NPY);  agouti-related protein (AgRP), cocaine- and amphetamine-regulated transcript (CART), and proopiomelanocortin (POMC). (Now line 40-42).

Point 4: Line 43: “Elevated leptin levels increase“ should be “Elevated leptin level increases”

RESPONSE 4: Thank you for this correction which we have incorporated in our revised MS. (Now line 47).

We greatly appreciate the corrections and suggestions made by the reviewers which have greatly improved our MS. 

Reviewer 2 Report

The manuscript is clear, relevant to the field and is presented in a well-structured manner.

The aim of the present study was to determine whether local administration of leptin into the commissural nucleus of the tractus solitarius alone, or after local anoxic stimulation with sodium cyanide (NaCN) in the isolated carotid sinus, modifies the glucose retention in the brain, expression of leptin Ob-Rb and C-Fos mRNA.

Authors conclude that leptin in the nucleus of the solitary tract potentiates brain glucose retention induced by anoxic stimuli within the isolated carotid body. But the most innovative result is that in this experimental situation (leptin in cNTS and anoxia within the isolated carotid body), the expression of the OB-Rb gene for leptin increases without showing an increase in neuronal activity as measured by c-Fos in cNTS.

Minnor specific comments:

  • Bibliography could be updated
  • In each figure legend you should indicate the statistical test you have performed.
  • In the section 2. Animals and general surgery:

In line 82-83, said:

“The experimental procedures were done in male Wistar rats weighing 270 to 300 g, 81 maintained individually in small acrylic cages with 12 light/dark cycles, and with free access to food and water with 10% glucose.”

Could you explain why do you administer water with glucose and not just water to the rats?

  • In the section 5. Isolated carotid sinus preparation, carotid chemoreceptors stimulation and BGR measurement.

In line 119, said: “vein The correct placement of all catheters was verified at the end of each experiment [13]. Could you explain briefly how you check the catheter placement? It is not clear what they mean by this phrase.

  • In Results section:
    • In line 216: …. (*p = 0.022), and when leptin + sal are compared (#p = 0.031). Can you explain why you put these values of p?
    • In line 218: “(Figure.3) b). But, when separate groups were compared with their respective times, this increase was significant only when aCSF + 219 NaCN at t = 10 min and 30 min were compared (&P = 0.043)” Remove the point between Figure and 3 and the b). The comparison made is not clear in this text.
    • In line 237: “however, this increase did not reach statistical significance (*p = 0.02)” There is a contradiction with what is written in the sentence and the value of p.
    • In line 240: please correct (aCSF + alen = 5)
    • In lines 246 y 271: please correct NacN and saliin respectively
  • In Discussion section:
    • In line 280: Remove the point between Figure and 5
    • In line 298: Bibliography could be updated

Author Response

The manuscript is clear, relevant to the field and is presented in a well-structured manner.

The aim of the present study was to determine whether local administration of leptin into the commissural nucleus of the tractus solitarius alone, or after local anoxic stimulation with sodium cyanide (NaCN) in the isolated carotid sinus, modifies the glucose retention in the brain, expression of leptin Ob-Rb and C-Fos mRNA.

Authors conclude that leptin in the nucleus of the solitary tract potentiates brain glucose retention induced by anoxic stimuli within the isolated carotid body. But the most innovative result is that in this experimental situation (leptin in cNTS and anoxia within the isolated carotid body), the expression of the OB-Rb gene for leptin increases without showing an increase in neuronal activity as measured by c-Fos in cNTS.

Minor specific comments:

Point 1. Bibliography could be updated 

Response 1: Thanks, the bibliography was updated. The next new references were added:

  1. Oberlin, D.; Buettner C. How does leptin restore euglycemia in insulin-deficient diabetes? J Clin Invest. 2017, 127, 450-453.
  2. Gao, L.; Ortega-Sáenz, P.; García-Fernández, M.; González-Rodríguez, P.; Caballero-Eraso, C.; López-Barneo, J. Glucose sensing by carotid body glomus cells: potential implications in disease. Frontiers in Physiology. 2014, 5, 1-9. 
  3. Maroni, P.; Citterio, L.; Piccoletti, R.; Bendinelli, P. Sam68 and ERKs regulate leptin-induced expression of OB-Rb mRNA in C2C12 myotubes. Mol Cell Endocrinol. 2009, 309, 26-31.
  4. Neyens, D.; Zhao, H.; Huston, N.J.; Wayman, G.A.; Ritter, R.C.; Appleyard, S.M. Leptin Sensitizes NTS Neurons to Vagal Input by Increasing Postsynaptic NMDA Receptor Currents. J Neurosci. 2020, 40, 7054-7064. 

5.- Akieda-Asai S, Poleni PE, Date Y. Coinjection of CCK and leptin reduces food intake via increased CART/TRH and reduced AMPK phosphorylation in the hypothalamus. Am J Physiol Endocrinol Metab. 2014, 306, E1284-291.

Point 2: In each figure legend you should indicate the statistical test you have performed.

Response 2: ANOVA with Scheffé post hoc test was used in all our statistical comparisons;  this is now indicated in each figure legend.

Point 3: In the section 2. Animals and general surgery:

In line 82-83, said:

“The experimental procedures were done in male Wistar rats weighing 270 to 300 g, 81 maintained individually in small acrylic cages with 12 light/dark cycles, and with free access to food and water with 10% glucose.”

Could you explain why do you administer water with glucose and not just water to the rats?

Response 3: In our experience, glucose levels are more stable during the experiment when  rats are maintained with 10% glucose 12 hours before the experiments.  When glucose was not present in the drinking water, we observed  a progressive decrease in basal glucose levels, likely due to a depletion of hepatic glycogen.

Point 4: In the section 5. Isolated carotid sinus preparation, carotid chemoreceptors stimulation and BGR measurement.

In line 119, said: “vein The correct placement of all catheters was verified at the end of each experiment [13]. Could you explain briefly how you check the catheter placement? It is not clear what they mean by this phrase.

Response 4:  This part has been clarified.  It was fundamentally the positioning of the placement of the jugular catheter that we were concerned about.  Since we access the jugular sinus through the jugular vein, we estimate the position of the catheter tip, by distance.  At the end of the experiments, we confirmed that the catheter tip was at the confluence of the external and internal jugular veins, therefore collecting blood from the entire encephalic circulation.    We have revised our description in the text to indicate:

The CS was isolated from systemic and cerebral circulations for 15 to 20 sec, during the time of application of NaCN. A polyethylene catheter (Clay Adams PE-10: 0.28 mm ID, 0.61 mm OD; Parsippany, NJ, USA) filled with heparin (1000 U/mL) was inserted via the lingual artery to the carotid (5 mm before the carotid bifurcation). NaCN (5µg/100g BW) or saline in 50 nL were injected through this catheter to the isolated carotid sinus. Any locally remaining NaCN was washed through the same syringe, before re-establishing normal circulation to the CS. To determine the BGR two catheters were inserted, one into the femoral artery, and the other in the jugular sinus from the external jugular vein. The correct placement of the jugular catheters was verified at the end of each experiment [13], by a neck skin incision to visualize the confluence of the external and internal jugular veins and the tip of the catheter. (Now line 126 and 127).

Point 5: In Results section:

In line 216: …. (*p = 0.022), and when leptin + sal are compared (#p = 0.031). Can you explain why you put these values of p?

Response 5:  We indicate the p values so that the reader can, without going to the figures, evaluate the statistical significance of our data.  However, we realize that the symbols in the text  (*,#, &) were confusing and we have eliminated them.  The specific p values are indicated in the legend to figure 3.

Now the text say: “The local application of NaCN to the ICS, in animals that received aCSF in cNTS (aCSF + NaCN, n = 10) resulted in a significant increase (p<0.05), compared with their basal values, in BGR at all time points studied.  A maximum increase was observed at t = 10 min post injection from 0.32 ± 0.08 to 1.3 ± 0.09 µmol/g/min. (Figure 3). This stimulation in BGR by NaCN, was significantly enhanced (p<0.001) by the local microinjection of leptin into cNTS (leptin + NaCN, n = 10), at all times studied.” (Now line 220-225).

Point 6: In line 218: “(Figure.3) b). But, when separate groups were compared with their respective times, this increase was significant only when aCSF + 219 NaCN at t = 10 min and 30 min were compared (&P = 0.043)” Remove the point between Figure and 3 and the b). The comparison made is not clear in this text.

Response 6:   Thank you for the correction.  The text now indicates: “But, when separate groups were compared with their respective times, this increase was significant only when Leptin + NaCN was compared with aCSF + NaCN at t = 10 min and 30 min (p = 0.044 and p = 0.038 respectively)”. (Now line 247-250)

Point 7:  In line 237: “however, this increase did not reach statistical significance (*p = 0.02)” There is a contradiction with what is written in the sentence and the value of p.

Response 7:   Thanks, the correction was made, now the p is 0.20. Sorry for the typographic error. (Now line 264).

Point 8:  In line 240: please correct (aCSF + alen = 5)

Response 8:   The text was corrected to: (aCSF + Sal = 5).  (Now line 267)

Point 9:  In lines 246 y 271: please correct NacN and saliin respectively

Response 9:   The text was corrected to: “NaCN” and “saline” respectively. (Now line 318 and 348 respectively)

Point 10:  In Discussion section:

In line 280: Remove the point between Figure and 5

Response 10:   Thank, the point was removed, now the text say: “Figure 5”.  Sorry for the typographic error. (Now line 357)

Point 11:  In line 298: Bibliography could be updated

 Response 11:   Thanks, the bibliography was updated: Gao, L.; Ortega-Sáenz, P.; García-Fernández, M.; González-Rodríguez, P.; Caballero-Eraso, C.; López-Barneo, J. Glucose sensing by carotid body glomus cells: potential implications in disease. Frontiers in Physiology. 2014, 5, 1-9.   (It is the number 15 in the bibliography).

Reviewer 3 Report

The study of the effects of leptin on the brain compartments expressing receptors to it - the commissural  nucleus  of  the  tractus  solitarius, C-fos without and with additional stimulation by hypoxia of chemoreceptors of the carotid body on glucose levels seems relevant. The paper shows that leptin enhances brain glucose retention through the expression of receptors to it in commissural  nucleus  of  the  tractus  solitaries.

In the captions to the figures, mean +SD should be given, since an abbreviation has already been given in the statistics section.

In the statistical analysis section, the abbreviation (SDM) should be deciphered and meas +/- SD should be written (line 189). It is not customary in this section to explain the symbols of the statistical significance of differences between groups - this is given in a note to the tables or figures (line 198-202). In this regard, it is desirable in the course of the presentation of the text to give the statistical significance of the differences between the groups without using symbols, and to indicate specifically between which groups they were identified (line 213-215, 218-220, 239, 255).

When interpreting the results of the effect of NaCN on the level of OB-Rb expression, the authors write about its increase, but further report that it is not statistically significant, and indicate in parentheses (*p=0.02) - is it statistically significant or not? (line 235-237)

Missing the opening parenthesis line 255.

In the captions to the figures, mean +/- SD should be given, since an abbreviation has already been given in the statistics section.

Author Response

The study of the effects of leptin on the brain compartments expressing receptors to it - the commissural  nucleus  of  the  tractus  solitarius, C-fos without and with additional stimulation by hypoxia of chemoreceptors of the carotid body on glucose levels seems relevant. The paper shows that leptin enhances brain glucose retention through the expression of receptors to it in commissural  nucleus  of  the  tractus  solitaries.

Point 1: In the captions to the figures, mean +SD should be given, since an abbreviation has already been given in the statistics section.

Response 1: Thanks, the abbreviation SDM has been put in the captions figures 3, 4 and 5.

Point 2: In the statistical analysis section, the abbreviation (SDM) should be deciphered and meas +/- SD should be written (line 189). It is not customary in this section to explain the symbols of the statistical significance of differences between groups - this is given in a note to the tables or figures (line 198-202). 

Response 2:  Thank you for this observation, the text corresponding to line 198-202 was eliminated from statistical analysis section: “When the basal mean values (at t -10 and t -5 min) for glucose levels, or BGR, were compared with other values, significance is represented by an asterisk symbol (*).  The hashtag symbol (#) indicates comparisons of the groups with leptin + saline group; the ampersand symbol (&) was used to compare the groups with aCSF + NaCN.”

Point 3: In this regard, it is desirable in the course of the presentation of the text to give the statistical significance of the differences between the groups without using symbols, and to indicate specifically between which groups they were identified (line 213-215, 218-220, 239, 255).

Response 3: Thank you for this suggestion, and now the text is: “The local application of NaCN to the ICS, in animals that received aCSF in cNTS (aCSF + NaCN, n = 10) resulted in a significant increase (p<0.05), compared with their basal values, in BGR at all time points studied.  A maximum increase was observed at t = 10 min post injection from 0.32 ± 0.08 to 1.3 ± 0.09 µmol/g/min. (Figure 3). This stimulation in BGR by NaCN, was significantly enhanced (p<0.001) by the local microinjection of leptin into cNTS (leptin + NaCN, n = 10), at all times studied.

These observations indicate that leptin in cNTS enhances BGR induced by ICS stimulation. Control rats group Leptin + Sal (n = 10), that received Sal in the ICS and leptin in the cNTS, did not show a significant change in BGR (Figure 3). But, when separate groups were compared with their respective times, this increase was significant only when Leptin + NaCN was compared with aCSF + NaCN at t = 10 min and 30 min (p = 0.044 and p = 0.038 respectively)”. (Now line 220-250).

Point 4: When interpreting the results of the effect of NaCN on the level of OB-Rb expression, the authors write about its increase, but further report that it is not statistically significant, and indicate in parentheses (*p=0.02) - is it statistically significant or not? (line 235-237).

Response 4:  Thank you for noting this mistake;  the p is 0.20, not 0.02;  this has been corrected. (Now line 264).

Point 5: Missing the opening parenthesis line 255.

Response 5:  The opening parenthesis has been inserted. (Now line 327).

Point 6: In the captions to the figures, mean +/- SD should be given, since an abbreviation has already been given in the statistics section.

Response 6:  As indicated above for reviewer #2 We now indicate for each legend the statistical test and include SDM in the captions to figures 3, 4 and 5.

We greatly appreciate the corrections and suggestions made by the reviewers which have greatly improved our MS. 
